# Synthesis, DFT Studies, Molecular Docking and Biological Activity Evaluation of Thiazole-Sulfonamide Derivatives as Potent Alzheimer’s Inhibitors

**DOI:** 10.3390/molecules28020559

**Published:** 2023-01-05

**Authors:** Shoaib Khan, Hayat Ullah, Muhammad Taha, Fazal Rahim, Maliha Sarfraz, Rashid Iqbal, Naveed Iqbal, Rafaqat Hussain, Syed Adnan Ali Shah, Khurshid Ayub, Marzough Aziz Albalawi, Mahmoud A. Abdelaziz, Fatema Suliman Alatawi, Khalid Mohammed Khan

**Affiliations:** 1Department of Chemistry, Hazara University, Mansehra 21120, Pakistan; 2Department of Chemistry, University of Okara, Okara 56130, Pakistan; 3Department of Clinical Pharmacy, Institute for Research and Medical Consultations (IRMC), Imam Abdulrahman Bin Faisal University, P.O. Box 1982, Dammam 31441, Saudi Arabia; 4Department of Zoology, Wildlife and Fisheries, University of Agriculture Faisalabad, Sub-Campus, Toba Tek Singh 36050, Pakistan; 5Department of Agronomy, Faculty of Agriculture and Environment, The Islamia University of Bahawalpur, Bahawalpur 63100, Pakistan; 6Department of Chemistry, University of Poonch, Rawalakot 12350, Pakistan; 7Faculty of Pharmacy, Universiti Teknologi MARA Cawangan Selangor Kampus Puncak Alam, Puncak Alam 42300, Selangor, Malaysia; 8Atta-ur-Rahman Institute for Natural Product Discovery (AuRIns), Universiti Teknologi MARA Cawangan Selangor Kampus Puncak Alam, Puncak Alam 42300, Selangor, Malaysia; 9Department of Chemistry, COMSATS University, Abbottabad Campus, Abbottabad 22060, Pakistan; 10Department of Chemistry, Alwajh College, University of Tabuk, Tabuk 71491, Saudi Arabia; 11Department of Chemistry, Faculty of Science, University of Tabuk, Tabuk 71491, Saudi Arabia; 12Department of Biochemistry, Faculty of Science, University of Tabuk, Tabuk 71491, Saudi Arabia; 13H.E.J. Research Institute of Chemistry, International Center for Chemical and Biological Sciences, University of Karachi, Karachi 75270, Pakistan

**Keywords:** synthesis, thiazole, sulfonamide, anti-Alzheimer’s, DFT, molecular docking

## Abstract

Alzheimer’s disease is a major public brain condition that has resulted in many deaths, as revealed by the World Health Organization (WHO). Conventional Alzheimer’s treatments such as chemotherapy, surgery, and radiotherapy are not very effective and are usually associated with several adverse effects. Therefore, it is necessary to find a new therapeutic approach that completely treats Alzheimer’s disease without many side effects. In this research project, we report the synthesis and biological activities of some new thiazole-bearing sulfonamide analogs (**1**–**21**) as potent anti-Alzheimer’s agents. Suitable characterization techniques were employed, and the density functional theory (DFT) computational approach, as well as in-silico molecular modeling, has been employed to assess the electronic properties and anti-Alzheimer’s potency of the analogs. All analogs exhibited a varied degree of inhibitory potential, but analog **1** was found to have excellent potency (IC_50_ = 0.10 ± 0.05 µM for AChE) and (IC_50_ = 0.20 ± 0.050 µM for BuChE) as compared to the reference drug donepezil (IC_50_ = 2.16 ± 0.12 µM and 4.5 ± 0.11 µM). The structure-activity relationship was established, and it mainly depends upon the nature, position, number, and electron-donating/-withdrawing effects of the substituent/s on the phenyl rings.

## 1. Introduction

Alzheimer’s disease (AD) is mainly related to the human brain. Hydrolysis of acetylcholine into choline acetic acid is the main activity of acetylcholinesterase (AchE) and butyrylcholinesterase (BuChE) enzymes [1]. Due to the hydrolysis effects, a shortage of acetylcholine products in the hippocampus and cortex of the brain is related to huge psychological functions [2]. Moreover, it is a continuous and irreversible brain disorder in which the cholinergic system of the brain is constantly imbalanced, often causing many consequences such as disorientation, difficulty in thinking, cognitive impairment, difficulty in problem solving, and memory loss [3,4,5]. For improvement in AD, the main focus is to target both AchE and BuChE enzymes [6,7]. Moreover, acetylcholinesterase is found in cholinergic neurons, muscle, and the brain, whereas butyrylcholinesterase is mainly present in the lungs, the heart, the liver, the kidneys, and the intestine [8,9,10]. Cleavage of an ester comprising analogs is due to the action and function of the enzyme AchE, which is dominant in the brain. Subsequently, when the functions of acetylcholine decrease, BuChE gradually increases. For this consideration, a potent drug is still required to diminish enzyme potentials [11]. In addition, the FDA approved various anti-Alzheimer’s drugs including donepezil, rivastigmine, tacrine, and galanthamine [12]. Furthermore, the limited use and applicability of drugs with insufficient activity cause gastrointestinal disturbance and hepatotoxicity [13,14,15,16].

Many biologically active compounds possessing a thiazole nucleus show significant potential and are considered one of the most widely used heterocyclic moieties [17]. The most important drug, penicillin, also contains a thiazole nucleus and demonstrates this basic fact and its significance [18]. Likewise, thiazole moiety is a core component of many bioactive drugs, such as Ravuconazole as an anti-fungal agent [19]; Dasatinib as an anti-neoplastic agent [20]; meloxicam and fentiazac as anti-inflammatory agents [21]; and nizatidine as an antiulcer agent [22], as shown in Figure 1.

Sulfonamide-containing compounds have received substantial attention in the last few decades and have emerged as potent inhibitors against various diseases, including diabetes [23], psychosis [24], central nervous system (CNS) disorders [25], tumors [26], and different cancer treatments [27]. 

Our research group had identified several heterocyclic compounds as potent therapeutics [28,29,30,31,32,33,34,35,36,37,38,39,40,41,42] and previously published thiazole and sulfonamide derivatives with various biological potentials [43,44,45,46] (see Figure 2). Keeping in view the biological importance of thiazole and sulfonamide derivatives, we have planned to design and synthesize a new hybrid class of thiazole-based sulfonamide analogs as acetylcholinesterase and butyrylcholinesterase inhibitors in search of lead candidates. 

## 2. Results and Discussion

### 2.1. Chemistry 

Different substituted sulfonyl chloride (**I**) was mixed with an excess of hydrazine hydrate in ethyl alcohol and refluxed for 5 h to give sulfonohydrazides (**II**) as the first intermediate product. The intermediate (**II**) was then treated with ammonium isothiocyanate in DMF under the refluxed condition to obtain the second intermediate product (**III**) [47]. The intermediate (**III**) was finally treated with different substituted phenacyl bromide in ethyl alcohol in the presence of triethylamine, and the mixture was refluxed for about 12 h to obtain thiazole-bearing sulfonamide analogs (**1**–**21**) (Figure 1, Table 1). After completion, the synthesized compounds were dried and then washed with *n*-hexane to obtain a pure product. Primary confirmation of the product was performed with the help of TLC, and further NMR confirmed the formation of the basic skeleton of the final products. 

A synthesized compound containing several heteroatoms along with carbon atoms has been described. These atoms are nitrogen, sulfur, oxygen, etc. Due to the presence of these atoms, protons (^1^H) and carbons (^13^C) were found with varied positions in ppm ranges. All the synthesized compounds (**1**–**21**) bearing different substituents on the varied position of the ring showed different peaks from one another. This might be due to the presence of electron-withdrawing and electron-donating groups. With the use of ^1^H NMR (500 MHz, DMSO), all the compounds having two protons directly attached with both the nitrogen atoms were found in the ranges between 11.0 and 11.80. Likewise, an aromatic singlet appeared at 11.67, and another singlet of a –OH proton appeared at 9.75. Similarly, the aromatic proton appeared at 8.29 showing doublet of doublet (dd) with coupling constant (*J*) = 7.4, 1.6 Hz. Furthermore, all other aromatic protons appeared with varied ranges, such as 7.60 showing dd with *J* = 1.2, 7.0 Hz; 7.59 showing dd with *J* = 1.8, 7.0 Hz; 7.22 showing singlet; 7.11 showing singlet; and 3.05 showing singlet, ^1^H, thiazole. Similarly, in the case of ^13^C NMR (125 MHz, DMSO): *δ,* all carbon appeared at varied ranges in descending order due to attached different substituents. The carbon appeared at 157.7, 148.4, 139.9, 132.9, 132.5, 131.4, 131.1, 130.4, 130.2, 129.2, 128.6, 128.5, 127.7, 125.2, and 122.0. 

### 2.2. Biological Activities

All the synthesized analogs of thiazole-bearing sulfonamide (**1**–**21**) were tested to explore their acetylcholinesterase (AchE) and butyrylcholinesterase inhibitory activities (see Figure 3).

#### 2.2.1. In Vitro Acetylcholinesterase (AchE) Inhibitory Activity 

All synthesized analogs showed good AchE inhibitory potentials with IC_50_ values ranging between 0.10 ± 0.05 and 11.40 ± 0.20 µM, as compared to the standard drug donepezil (IC_50_ = 2.16 ± 0.12 µM) (Table 1). The structure–activity relationship was carried out for all the analogs, which mainly depend upon the nature, number, and position of the substituent/s on the phenyl ring. 

If we compare analog **3** (IC_50_ = 2.90 ± 0.10 µM) with analog **18** (IC_50_ = 0.80 ± 0.050 µM) and **20** (IC_50_ = 3.10 ± 0.10 µM), all analogs have the same nitro groups on phenyl rings A and B. Differences in the activities of the entire analog may be due to the different numbers and positions of the nitro group (see Figure 4). 

Similarly, we compare analog **7** (IC_50_ = 2.60 ± 0.10 µM) with analog **15** (IC_50_ = 0.40 ± 0.10 µM). Both analogs have chloro groups on phenyl rings A and B. The small difference in the inhibitory potentials of this analog may be due to the different number and position of the same chloro substituent on phenyl rings A and B (see Figure 5). 

Comparing analog **8** (IC_50_ = 7.30 ± 0.10) with analog **16** (IC_50_ = 0.30 ± 0.050 µM), **19** (IC_50_ = 3.10 ± 0.10 µM), and **21** (IC_50_ = 3.20 ± 0.10 µM) reveals that all analogs comprise a phenyl ring attached to the *para*-position of ring B, but ring A possesses a different type of substituents at different positions, i.e., analog **8** contains a chloro group at the *para*-position, analog **16** contains nitro at *meta* and methyl at the *ortho*-position, analog **19** contains two nitro groups at the *ortho*- and *meta*-position, and analog **21** contains a nitro group at the *para*-position. The difference in the potentials of this analog may be due to the presence of different types and positions of the substituents on phenyl ring A (see Figure 6). 

#### 2.2.2. In Vitro Butyrylcholinesterase (BchE) Inhibitory Activity

All synthesized analogs showed good BuChE inhibitory potentials with IC_50_ values ranging between 0.20 ± 0.050 and 14.30 ± 0.30 µM as compared to the standard drug donepezil (IC_50_ = 4.5 ± 0.11 µM) (Table 1).

Comparing analog **16** (IC_50_ = 0.20 ± 0.050 µM) with analog **21** (IC_50_ = 3.20 ± 0.10 µM) reveals that both analogs have a phenyl ring attached to the para-position of the ring B, and ring A has nitro and methyl groups in analog **16** and only nitro groups in analog **21**. The difference in the activity may be due to the different nature and position of the group attached to ring A (see Figure 7). 

Similarly, if we compare analog **1** (IC_50_ = 0.10 ± 0.05 µM) with analog **2** (IC_50_ = 1.90 ± 0.10 µM) and **10** (IC_50_ = 0.60 ± 0.050), analogs **1** and **2** both have a nitro group on ring B but an analog **10**-bearing nitro group on ring A. The difference in their biological potential might be due to the position of substituents on the rings. The potential difference of analog **10** being somewhat lower than analog **1** may be due to the attachment of methyl moiety at the *ortho*-position of ring A, which produces steric hindrance; therefore, the activity of the analog was found to be lower (see Figure 8). 

The nature of substituents, the number of substituents, and the position at which substituents are attached might increase or decrease the biological potential of the analog. The nature of attached substituents such as an electron donating group (EDG) increases the biological potential due to the transfer of an electron, which causes a negative charge in the ring as well as the formation of a hydrogen bond with enzyme active sites. However, in the case of an electron-withdrawing group (EWG), the nature of the attached substituents decreases the biological potential. In this consideration, it was concluded that the number, position, and nature of the substituents can increase or decrease the biological potentials of the analog. The binding interaction of most analogs with the active site of enzymes was determined with the help of molecular docking. 

### 2.3. Docking Study

The molecular docking study was carried out to gain insight into the binding mode of synthesized compounds against both the targeted enzyme. The optimized compounds were docked into the active site of targeted enzymes based on the co-crystal of each crystallographic structure. A total of nine poses were obtained for each ligand molecule in which the top-ranked was selected to explore the binding sites using the discovery studio visualizer (DSV). The docking results revealed that all the compounds were found in a good orientation in the active site of both enzymes. Generally, we have noticed that all the compounds hold different substituting groups, i.e., electron-withdrawing (also known as deactivated) and electron-donating groups at different positions. Interestingly, the protein–ligand interaction (PLI) profile with the in vitro results revealed that analogs **1** and **10** showed the best potential against both compounds and had been ranked 1st and 2nd in the series of all compounds. Both of the compounds bear a strong magnitude of activated and deactivated groups over the rings. The detailed PLI profile of both compounds against both targets revealed numerous key interactions with catalytic residues, which might have a potential role in the enhancement of the enzymatic activity of both enzymes. 

For the molecular docking study, both proteins (1Acl and 1PoP) were retrieved from the RCSD protein data bank (PDB) and performed using different software such as Auto Dock Vina (1.5.7) and DSV (2021). Initially, proteins were prepared in DSV by removing water and transferring data in the PDB format to Auto Dock Vina, where polar hydrogen and charges were added. Next, ligand molecules were added, and X, Y, and Z coordinates were saved in text format; however, both protein and ligand were also saved in PDBQT format. The location was set in the command prompt through which all the docking procedure was carried out. All the results were decoded in DSV, and the interactions were visualized in both 2D and 3D structures (see Figure 9, Figure 10, Figure 11 and Figure 12). Their protein–ligand interactions with distance are summarized in Table 2.

The comparison studies of donepezil (see Figure 13 and Figure 14) with synthesized compounds were obtained through molecular docking studies and their binding interaction with active sites of protein (PLI). Synthesized compounds showed much better interaction as compared to the standard drug donepezil; the interaction is summarized in Table 2. 

### 2.4. Computational Details

All the DFT calculations for thiazole-bearing sulfonamide analogs (**1**–**21**) are computed using the Gaussian 09 quantum chemical package [48]. The geometries of considered thiazole-bearing sulfonamide analogs (**1**–**21**) are optimized at ωB97XD/6–31 g (d,p) level of theory. ωB97XD functional is used due to the associated accuracy and validity of the results for geometric as well as electronic parameters [49,50]. Additionally, vibrational frequency analysis is also carried out to confirm the true minimum nature of the optimized geometries (**1**–**21**) on the potential energy surface (PES). Frontier molecular orbital (FMO) analysis is also performed at the same level of theory to understand the perturbations in electronic properties. Moreover, molecular electrostatic potential (MESP) is also extracted to gain insight into the interaction mode of designed analogs with the targeted enzymes. The Chemcraft package and Gauss View 5.0 are employed for the visualization of geometries and isodensity [51].

The absence of negative or imaginary frequencies in all the designed thiazole-bearing sulfonamide analogs (**1**–**21**) confirms the true minima nature of the stationary points on PES. The optimized structures of analogs **1** and **2** are shown in Figure 13, whereas the optimized geometries of analogs (**3**–**21**) are presented in Appendix A. Because all thiazole derivatives are almost similar in structure, the same geometric parameters are used to characterize their geometries. Slight differences of ±0.01Å in bond lengths (S—C and S—N) are observed for analogs **1** and **2,** which are attributed to different functionalization of ring **A** (see Figure 15). The angle ∠N—N—S at the sulfonamide moiety is 116^o^ in the case of analog **1**, whereas in analog **2,** this angle is slightly decreased to 115°. The N—S bond is 1.68Å and 1.69Å in analogs **1** and **2**, respectively. Similarly, the connecting bond of sulfonamide moiety and ring **A** (S—C bond) also differs by 0.01Å in both analogs. However, the C—C bond between thiazole moiety and ring **B** is similar in both analogs due to structural similarity at this end. An almost similar geometric pattern is observed in other optimized analogs (**3**–**21**) given in Appendix A. For comparison, the optimized structure of the standard drug (donepezil) is also added in Figure 15.

### 2.5. Molecular Electrostatic Potential 

Molecular electrostatic potential (MESP) is a very useful tool that predicts the relationships of physicochemical properties of designed drug molecules [52]. MESP is also important for validating the reactivity of the drug molecules toward nucleophilic or electrophilic attacks [53]. Moreover, MESP mapping also provides a valuable reference for the interaction of drug molecules with the targeted enzyme by evaluating electrostatic interactions [54]. The MESP mapping of thiazole-bearing sulfonamide analogs **1** and **2** is shown in Figure 16, whereas for analogs (**3**–**21**), MESP maps are presented in Appendix A. In the MESP, the higher negative region (red color) is the favorite site for electrophilic attack. Therefore, electrophiles will be more likely to attack nucleophilic sites, and the opposite is true for the blue color regions (positive potential). 

The MESP map of analog **1** reveals that the oxygen atoms of the NO_2_ group (attached to ring B) and sulfonamide have a negative potential region (orange color). On the other hand, hydrogen atoms attached to the nitrogen of sulfonamide and the OH^−^ group of ring A exhibit the blue color corresponding to a maximum positive charge. The most positive potential hydrogen atoms ascribe the polar nature of N—H and O—H bonds. However, the MESP over chlorine atoms seems to have nearly neutral electrostatic potential due to lower negative potential over them as compared to O atoms. Similar results for the negative potential of MESP are observed for analog **2**; however, positive density (blue region) is distributed only over hydrogen atoms of sulfonamide attributed to the higher polarity of the N—H bond. In the remaining studied analogs (**3**–**21**), similar positive potential regions (blue color) are observed in the MESP, and these regions correspond to hydrogen atoms attached to the nitrogen of sulfonamide. However, the negative potential regions (reddish color) vary depending on the position of NO_2_ groups (see Appendix A for more details). Similarly, the MESP of the standard drug (donepezil) is also calculated for comparison with the studied analogs. In the case of the standard drug, the negative potential region (red color) is distributed over the oxygen atom of cyclopentenone. However, positive density (blue region) is mainly distributed over hydrogen atoms of the methoxy group.

### 2.6. Frontier Molecular Orbital Analysis

The frontier molecular orbitals, HOMO and LUMO, help determine the chemical reactivity of a molecule when it interacts with the target enzyme or protein. The HOMO–LUMO gap helps to characterize the kinetic stability and chemical reactivity of the molecule [55]. Therefore, frontier molecular orbital (FMO) analysis is performed to gain insight into the electronic properties of the designed thiazole-bearing sulfonamide analogs (**1**–**21**). HOMO–LUMO orbital densities of analogs 1 and 2 and the standard drug are presented in Figure 15, whereas the HOMO–LUMO isodensity for studied thiazole-bearing sulfonamide analogs (**1**–**21**) are shown in Appendix A. Results of HOMO–LUMO energies and their energy gaps are reported in Table 3. In the case of sulfonamide analogs **1** and **2**, almost similar behavior is observed regarding HOMO–LUMO densities. In both analogs, the HOMOs are mainly distributed on ring **B** and the thiazole ring, with some density over N atoms of sulfonamide as well (see Figure 17). On the other hand, LUMO is distributed entirely on ring **B** in both analogs **1** and **2**. The distribution pattern of HOMO densities of analogs (**3**–**21**) is consistent with analogs **1** and **2.** However, the LUMO densities of analogs (**3**–**21**) are quite different and distributed over ring **A** (see Appendix A). Thus, the charge is transferred from ring **B** and the thiazole ring to ring **A**. The FMO density distribution pattern is similar in all remaining analogs (**3**–**21**), in which LUMO is mainly distributed over ring **A** and HOMO over ring **B** and the thiazole moiety (see Appendix A). 

Analog **1** has an energy gap of 7.91 eV, and analog **2** has an energy gap of 7.92 eV. However, the HOMO and LUMO values of analog **1** are −8.36 eV and −0.46 eV, respectively. A higher HOMO–LUMO energy gap renders the higher electronic stability of thiazole-bearing sulfonamide analogs. Overall, the highest energy gap is calculated for analog **15** (−8.38 eV), whereas the lowest HOMO–LUMO gap is seen in analog **21** (−6.62 eV). The higher HOMO values are associated with the electron-donating ability of the designed molecules, whereas the lower LUMO energies render good electron-accepting ability [56]. The HOMO–LUMO energy values indicate that the charge transfer occurs within the thiazole-bearing sulfonamide analogs. Therefore, results in Table 3 reveal that lower LUMO energy values promote the electron-accepting ability of considered analogs (**1**–**21**) and higher HOMO values correspond to the electron-donating tendency of analogs. FMO analysis of a standard drug (donepezil) shows almost similar results. HOMO, LUMO, and the energy gap values of the standard drug are almost consistent with the studied analogs, especially with analogs 5, 7, and 15 (see Table 3). Moreover, the distribution of HOMO–LUMO isodensity also confirms that charge transfer occurs within the drug molecule, which is in good correlation with studied thiazole-bearing sulfonamide analogs.

## 3. Conclusions

In summary, we have synthesized twenty-one analogs of thiazole-bearing sulfonamide and evaluated them against acetylcholinesterase and butyrylcholinesterase in the presence of the standard drug donepezil (IC_50_ values = 2.16 ± 0.12 and 4.5 ± 0.11 µM, respectively). All analogs were found with different AchE and BuChE inhibitory activity, having IC_50_ values ranging between 0.10 ± 0.05 and 11.40 ± 0.20 µM and between 0.20 ± 0.050 and 14.30 ± 0.30 µM, respectively. Analog **1** was found to be the most potent one in both AchE and BuChE cases with IC_50_ values = 0.10 ± 0.05 µM and 0.20 ± 0.050 µM, respectively. A limited structure–activity relationship was carried out to find the effect of different substituents on phenyl rings A and B. Molecular docking studies were carried out to find the interaction of the most potent analog with the active site of enzymes. Moreover, DFT was conducted to determine the chemical reactivity of a molecule when it interacts with the target enzyme or protein. 

## 4. Experimental

### 4.1. General Procedure for the Synthesis of Thiazole-Bearing Sulfonamide Analogs (**1**–**21**)

Thiazole-bearing sulfonamide derivatives (**1**–**21**) were obtained by treating different substituted sulfonyl chloride (**I,** 1 mmol) with hydrazine hydrate (15 mL) in ethyl alcohol (10 mL) and refluxed for 5 h to give sulfonohydrazides (**II**) as a first intermediate product. The intermediate (**II**) was then treated with equimolar ammonium isothiocyanate in DMF (10 mL) to obtain the second intermediate product (**III**) [40]. The intermediate (**III**) was finally treated with equimolar different substituted phenacyl bromide in ethyl alcohol (10 mL) in the presence of triethylamine and refluxed for about 12 h to obtain the desired product. After completion, the synthesized compound was dried and then washed with *n*-hexane to obtain a pure product. Primary confirmation of the product was performed with the help of TLC, and further NMR confirms the formation of the basic skeleton of the final products. 

### 4.2. Spectral Analysis

#### 4.2.1. 3,5-Dichlro-2-hydroxy-N′-(4-(3-nitrophenyl)thiazol-2-yl)benzenesulfonohydrazide (**1**)

Yield: 82%; m.p: 209–211 °C. ^1^H NMR (500 MHz, DMSO-*d_6_): δ* 11.97 (s, 1H, NH), 11.92 (s, 1H, NH), 10.45 (s, 1H, OH), 8.88 (s, 1H, Ar-H), 8.81 (s, 1H, Ar-H), 8.57 (s, 1H, Ar-H), 7.59 (t, *J* = 6.7 Hz, 1H, Ar-H), 6.78 (dd, *J* = 7.2, 1.8 Hz, 1H, Ar-H), 6.61 (dd, *J* = 7.3, 2.0 Hz, 1H, Ar-H), 3.18 (s, 1H, thiazole-H). ^13^C NMR (125 MHz, DMSO-*d_6_)*: *δ* 162.6, 150.2, 149.8, 149.7, 142.6, 142.4, 127.9, 127.7, 120.8, 118.9, 118.8, 118.7, 117.0, 114.1, 100.9. HR EIMS: *m*/*z* calcd for C_15_H_10_Cl_2_N_4_O_5_S_2_ [M]^+^ 461.2978; Found: 461.2960.

#### 4.2.2. 4-Chloro-N′-(4-(3-nitrophenyl)thiazol-2-yl)benzenesulfonohydrazide (**2**)

Yield: 86%.; m.p: 207–210 °C. ^1^H NMR (500 MHz, DMSO-*d_6_): δ* 12.15 (s, 1H, NH), 11.92 (s, 1H, NH), 8.79 (d, *J* = 7.7 Hz, 2H, Ar-H), 8.51 (d, *J* = 6.7 Hz, 2H, Ar-H), 8.43 (s, 1H, Ar-H), 7.58 (d, *J* = 8.5, 2.4 Hz, 1H, Ar-H), 7.35 (t, *J* = 6.6 Hz, 1H, Ar-H), 6.61 (dd, *J* = 7.3, 1.4 Hz, 1H, Ar-H), 2.36 (s, 1H, CH). ^13^C NMR (125 MHz, DMSO-*d_6_)*: *δ* 162.0, 161.4, 159.6, 149.6, 146.4, 142.5, 127.7, 127.6, 126.4, 121.0, 120.7, 118.7, 100.9, 99.1, 94.4. HR EIMS: *m*/*z* calcd for C_15_H_11_ClN_4_O_4_S_2_ [M]^+^ 410.8552; Found: 410.8540.

#### 4.2.3. 3-Nitro-N′-(4-(3-nitrophenyl)thiazol-2-yl)benzenesulfonohydrazide (**3**)

Yield: 80%; m.p: 221–225 °C. ^1^H NMR (500 MHz, DMSO-*d_6_): δ* 11.90 (s, 1H, NH), 11.78 (s, 1H, NH), 8.81 (s, 1H, Ar-H), 8.77 (s, 1H, Ar-H), 8.53 (s, 1H, Ar-H), 7.85 (s, 1H, Ar-H), 7.58 (d, *J* = 6.8 Hz, 1H, Ar-H), 7.31-7.29 (m, 1H, Ar-H), 6.61 (d, *J* = 7.4 Hz, 1H, Ar-H), 7.28–7.25 (m, 1H, Ar-H), 2.46 (s, 1H, CH). ^13^C NMR (125 MHz, DMSO-*d_6_)*: *δ* 162.8, 149.7, 149.6, 147.5, 139.7, 131.7, 129.3, 127.6, 126.9, 121.3, 118.7, 100.9. HR EIMS: *m*/*z* calcd for C_15_H_11_N_5_O_6_S_2_ [M]^+^ 421.4032; Found: 421.4018.

#### 4.2.4. 2-Methyl-5-nitro-N′-(4-(3-nitrophenyl)thiazol-2-yl)benzenesulfonohydrazide (**4**)

Yield: 77%; m.p: 203–205 °C. ^1^H NMR (500 MHz, DMSO-*d_6_): δ* 11.40 (s, 1H, NH), 11.27 (s, 1H, NH), 8.64 (d, *J* = 2.1 Hz, 1H, Ar-H), 8.56 (d, *J* = 1.9 Hz, 1H, Ar-H), 8.43 (d, *J* = 1.8 Hz, 1H, Ar-H), 8.29 (dd, *J* = 7.7, 1.8 Hz, 1H, Ar-H), 8.19 (m, 1H, Ar-H), 7.79 (dd, *J* = 7.7, 1.9 Hz, 1H, Ar-H), 7.63 (d, *J* = 1.9 Hz, 1H, Ar-H), 2.12 (s, 1H, CH), 2.60 (s, 3H, CH_3_). ^13^C NMR (125 MHz, DMSO-*d_6_)*: *δ* 173.0, 150.0, 148.1, 145.1, 142.3, 139.7, 133.6, 133.6, 130.5, 130.1, 126.9, 123.9, 123.0, 122.5, 105.0, 22.0. HR EIMS: *m*/*z* calcd for C_16_H_13_N_5_O_6_S_2_ [M]^+^ 435.4345; Found: 435.4330.

#### 4.2.5. 4-Bromo-N′-(4-phenylthiazol-2-yl)benzenesulfonohydrazide (**5**)

Yield: 87%; m.p: 198–201 °C. ^1^H NMR (500 MHz, DMSO-*d_6_): δ* 11.46 (s, 1H, NH), 11.25 (s, 1H, NH), 7.85 (d, *J* = 2.1 Hz, 2H, Ar-H), 8.82(d, *J* = 1.9 Hz, 2H, Ar-H), 7.80 (dd, *J* = 7.5, 1.8 Hz, 2H, Ar-H), 7.45 (m, 1H, Ar-H), 7.35 (dd, *J* = 7.1, 1.9 Hz, 2H, Ar-H), 2.09 (s, 1H, CH), ^13^C NMR (125 MHz, DMSO-*d_6_)*: *δ* 173.1, 150.1, 137.6, 133.0, 131.8, 131.7, 129.4, 129.3, 129.1, 129.0, 128.6, 127.4, 127.3, 126.2, 105.0. HR EIMS: *m*/*z* calcd for C_15_H_12_BrN_3_O_2_S_2_ [M]^+^ 410.3080; Found: 410.3075, 412.3076.

#### 4.2.6. 4-Nitro-N′-(4-phenylthiazol-2-yl)benzenesulfonohydrazide (**6**)

Yield: 85%; m.p: 211–214 °C. ^1^H NMR (500 MHz, DMSO-*d_6_): δ* 11.83 (s, 1H, NH), 11.64 (s, 1H, NH), 8.35 (d, *J* = 7.9 Hz, 2H, Ar-H), 7.62 (d, *J* = 7.8 Hz, 2H, Ar-H), 7.80 (dd, *J* = 7.7, 1.9 Hz, 2H, Ar-H), 7.45 (m, 1H, Ar-H), 7.35 (dd, *J* = 7.1 Hz, 2H, Ar-H), 4.12 (s, 1H, CH), ^13^C NMR (125 MHz, DMSO-*d_6_)*: *δ* 173.0, 151.0, 150.2, 142.7, 133.0, 129.1, 128.9, 128.6, 128.1, 128.0, 127.4, 127.3, 124.1, 124.0, 105.0. HR EIMS: *m*/*z* calcd for C_15_H_12_N_4_O_4_S_2_ [M]^+^ 376.4180; Found: 376.4155.

#### 4.2.7. 4-Chloro-N′-(4-(4-chlorophenyl)thiazol-2-yl)benzenesulfonohydrazide (**7**)

Yield: 80%; m.p: 206–208 °C. ^1^H NMR (500 MHz, DMSO-*d_6_): δ* 11.20 (s, 1H, NH), 11.14 (s, 1H, NH), 7.62 (d, *J* = 6.3 Hz, 2H, Ar-H), 7.59 (d, *J* = 8.5 Hz, 2H, Ar-H), 7.41 (d, *J* = 6.3 Hz, 2H, Ar-H), 7.39 (d, *J* = 6.2 Hz, 2H, Ar-H), 7.29 (s, 1H, CH). ^13^C NMR (125 MHz, DMSO-*d_6_): δ* 157.6, 154.3, 148.8, 146.6, 144.4, 136.2, 133.2, 131.1, 129.7, 129.6, 129.0, 128.8, 128.2, 127.7, 127.6. HR EIMS: *m*/*z* calcd for C_15_H_11_C_l2_N_3_O_2_S_2_ [M]^+^ 400.2940; Found: 400.2925. 

##### 4.2.8. N′-(4-([1,1′-biphenyl]-4-yl)thiazol-2-yl)-4-chlorobenzenesulfonohydrazide (**8**)

Yield: 75%; m.p: 168–174 °C. ^1^H NMR (500 MHz, DMSO-*d_6_): δ* 11.65 (s, 1H, NH), 11.41 (s, 1H, NH), 8.22 (d, *J* = 7.7 Hz, 2H, Ar-H), 7.81 (d, *J* = 7.5 Hz, 2H, Ar-H), 7.73 (dd, *J* = 7.5, 1.9 Hz, 2H, Ar-H), 7.69 (d, *J* = 7.5 Hz, 2H, Ar-H), 7.56 (d, *J* = 7.1 Hz, 2H, Ar-H), 7.43 (dd, *J* = 7.1 Hz, 2H, Ar-H), 7.35 (m, *J* = 7.1–7.31H, Ar-H), 3.25 (s, 1H, CH). ^13^C NMR (125 MHz, DMSO-*d_6_): δ* 173.1, 150.2, 140.5, 140.3, 137.1, 134.4, 131.2, 129.1, 128.7, 128.7, 128.4, 128.2, 128.1, 127.5, 127.6, 127.4, 127.3, 127.1, 127.1, 127.0, 105.1. HR EIMS: *m*/*z* calcd for C_21_H_16_ClN_3_O_2_S_2_ [M]^+^ 441.9523; Found: 441.9510.

##### 4.2.9. 3,5-Dichloro-N′-(4-(4-chlorophenyl)thiazol-2-yl)-2-hydroxybenzenesulfonohydrazide (**9**)

Yield: 78%; m.p: 191–196 °C. ^1^H NMR (500 MHz, DMSO-*d_6_): δ* 11.88 (s, 1H, NH), 11.67 (s, 1H, NH), 9.35 (s, 1H, OH), 8.78 (d, *J* = 1.8 Hz, 1H, Ar-H), 8.50 (d, *J* = 1.7 Hz, 1H, Ar-H), 7.57 (d, *J* = 6.7 Hz, 2H, Ar-H), 6.29 (d, *J* = 6.7 Hz, 2H, Ar-H), 3.20 (s, 1H, CH). ^13^C NMR (125 MHz, DMSO-*d_6_): δ* 162.6, 149.7, 146.1, 145.5, 142.6, 142.7, 127.9, 127.8, 120.6, 120.1, 120.0, 119.0, 118.7, 117.3, 101.0. HR EIMS: *m*/*z* calcd for C_15_H_10_Cl_3_N_3_O_3_S_2_ [M]^+^ 450.7385; Found: 450.7345.

##### 4.2.10. N′-(4-(3,4-dichlorophenyl)thiazol-2-yl)-2-methyl-5-nitrobenzenesulfonohydrazide (**10**)

Yield: 72%; m.p: 187–191 °C. ^1^H NMR (500 MHz, DMSO-*d_6_): δ* 11.70 (s, 1H, NH), 11.59 (s, 1H, NH), 8.50 (s, 1H, Ar-H), 8.49 (d, 1H, *J* = 7.8 Hz, Ar-H), 8.07 (dd, *J* = 8.2, 2.6 Hz, 1H, Ar-H), 7.98 (d, *J* = 8.5 Hz, 1H, Ar-H), 7.94 (d, *J* = 8.5 Hz, 1H, Ar-H), 7.70 (d, *J* = 6.6 Hz, 1H, Ar-H), 2.70 (s, 1H, CH), 2.63 (s, 3H, CH_3_). ^13^C NMR (125 MHz, DMSO-*d_6_): δ* 165.3, 162.8, 160.6, 158.2, 147.1, 144.8, 144.2, 132.8, 132.5, 132.2, 132.1, 132.1, 132.1, 131.8, 131.8. HR EIMS: *m*/*z* calcd for C_16_H_12_C_l2_N_4_O_4_S_2_ [M]^+^ 459.3285; Found: 459.3278.

##### 4.2.11. N′-(4-(4-chlorophenyl)thiazol-2-yl)-2-methyl-5-nitrobenzenesulfonohydrazide (**11**)

Yield: 84%; m.p: 213–216 °C. ^1^H NMR (500 MHz, DMSO-*d_6_): δ* 11.69 (s, 1H, NH), 11.56 (s, 1H, NH), 8.01-8.10 (m, 4H, Ar-H), 7.58-7.60 (m, 7.3–7.51H, Ar-H), 7.47 (d, *J* = 8.2 Hz, 1H, Ar-H), 3.11 (s, 1H, CH) 2.65 (s, 3H, CH_3_). ^13^C-NMR (125 MHz, DMSO-*d_6_): δ* 173.1, 152.1, 145.1, 142.6, 139.7, 134.7, 131.7, 130.2, 129.3, 129.2, 128.5, 128.5, 127.1, 123.2, 104.9, 22.1. HR EIMS: *m*/*z* calcd for C_16_H_13_ClN_4_O_4_S_2_ [M]^+^ 424.8723; Found: 424.8707.

##### 4.2.12. N′-(4-(4-bromophenyl)thiazol-2-yl)-2-methyl-5-nitrobenzenesulfonohydrazide (**12**)

Yield: 74%; m.p: 195–199 °C; ^1^H NMR (500 MHz, DMSO-*d_6_): δ* 11.70 (s, 1H, NH), 11.54 (s, 1H, NH), 8.51 (d, *J* = 2.5 Hz, 1H, Ar-H), 8.10 (dd, *J* = 7.4, 2.6 Hz, 1H, Ar-H), 7.95 (d, *J* = 8.4 Hz, 2H, Ar-H), 7.75 (d, *J* = 8.6 Hz, 2H, Ar-H), 7.47 (d, *J* = 8.2 Hz, 1H, Ar-H), 4.06 (s, 1H, CH), 2.65 (s, 3H, CH_3_). ^13^C NMR (125 MHz, DMSO-*d_6_): δ* 165.4, 158.4, 147.6, 144.8, 144.2, 132.6, 132.2, 132.1, 131.9, 131.6, 131.5, 131.4, 131.3, 131.2, 131.1, 128.5. HR EIMS: *m*/*z* calcd for C_16_H_13_BrN_4_O_4_S_2_ [M]^+^ 469.3398; Found: 469.3385, 471.3384.

##### 4.2.13. N′-(4-(3,4-dichlorophenyl)thiazol-2-yl)-3-nitrobenzenesulfonohydrazide (**13**)

Yield: 77%; m.p: 186–189 °C. ^1^H NMR (500 MHz, DMSO-*d_6_): δ* 11.88 (s, 1H, NH), 11.76 (s, 1H, NH), 8.34 (t, 1H, Ar-H), 8.25 (d, *J* = 1.9 Hz, 1H, Ar-H), 8.03 (dd, *J* = 7.6, 2.4 Hz, 1H, Ar-H), 8.00 (dd, *J* = 7.9, 3.9 Hz, 1H, Ar-H), 7.80 (d, *J* = 8.3 Hz, 1H, Ar-H), 7.80 (dd, *J* = 7.4, 1.7 Hz, 1H, Ar-H), 7.6 (t, *J* = 7.4 Hz, 1H, Ar-H), 4.02 (s, 1H, CH). ^13^C NMR (125 MHz, DMSO-*d_6_): δ* 164.2, 149.7, 147.1, 135.9, 133.6, 131.3, 131.2, 130.8, 129.7, 129.5, 129.3, 127.7, 123.4, 120.0, 89.9. HR EIMS: *m*/*z* calcd for C_15_H_10_Cl_2_N_4_O_4_S_2_ [M]^+^ 445.2945; Found: 445.2931.

##### 4.2.14. N′-(4-(4-chlorophenyl)thiazol-2-yl)-3-nitrobenzenesulfonohydrazide (**14**)

Yield: 85; m.p: 166–172 °C. ^1^H NMR (500 MHz, DMSO-*d_6_): δ* 11.90 (s, 1H, NH), 11.80 (s, 1H, NH), 8.79 (d, *J* = 6.7 Hz, 2H, Ar-H), 8.52 (d, *J* = 7.7 Hz, 2H, Ar-H), 8.45 (s, 1H, Ar-H), 7.58 (t, *J* = 6.7 Hz, 1H, Ar-H), 7.02 (dd, *J* = 7.9, 1.4 Hz, 1H, Ar-H), 6.61 (dd, *J* = 7.8, 1.4 Hz, 1H, Ar-H), 3.81 (s, 1H, CH). ^13^C NMR (125 MHz, DMSO-*d_6_): δ* 159.6, 149.6, 147.3, 142.7, 135.8, 129.8, 127.6, 126.4, 121.2, 1120.5, 19.9, 118.7, 116.0, 111.3, 100.9. HR EIMS: *m*/*z* calcd for C_15_H_11_ClN_4_O_4_S_2_ [M]^+^ 410.8556; Found: 410.8540. 

##### 4.2.15. 4-Chloro-N′-(4-(3,4-dichlorophenyl)thiazol-2-yl)benzenesulfonohydrazide (**15**)

Yield: 79%; m.p: 193–197 °C. ^1^H NMR (500 MHz, DMSO-*d_6_): δ* 11.90 (s, 1H, NH), 11.51 (s, 1H, NH), 8.51 (s, 1H, Ar-H), 7.58 (dd, *J* = 7.2, 1.5 Hz, 1H, Ar-H), 7.29 (d, *J* = 7.0 Hz, 1H, Ar-H), 6.60 (d, *J* = 6.8 Hz, 2H, Ar-H), 6.38 (d, *J* = 6.9 Hz, 2H, Ar-H), 3.18 (s, 1H, CH), ^13^C NMR (125 MHz, DMSO-*d_6_): δ* 162.3, 160.6, 159.5, 149.7, 149.0, 142.5, 131.3, 127.8, 127.7, 120.8, 118.7, 110.5, 107.6, 102.7, 100.9. HR EIMS: *m*/*z* calcd for C_15_H_10_Cl_3_N_3_O_2_S_2_ [M]^+^ 434.7312; Found: 434.7301.

##### 4.2.16. N′-(4-([1,1′-biphenyl]-4-yl)thiazol-2-yl)-2-methyl-5-nitrobenzenesulfonohydrazide (**16**)

Yield: 72%; m.p: 173–175 °C. ^1^H NMR (500 MHz, DMSO-*d_6_): δ* 11.89 (s, 1H, NH), 11.87 (s, 1H, NH), 8.79 (d, *J* = 7.6 Hz, 1H, Ar-H), 8.52 (d, *J* = 6.7 Hz, 1H, Ar-H), 7.89 (s, 1H, Ar-H), 7.57 (d, *J* = 6.9 Hz, 2H, Ar-H), 7.11 (d, *J* = 7.9 Hz, 2H, Ar-H), 6.60 (dd, *J* = 7.3, 1.5 Hz, 2H, Ar-H), 7.42–7.39 (m, 2H, Ar-H), 7.02 (t, *J* = 7.2 Hz, 1H, Ar-H), 3.88 (s, 1H, CH), 1.91 (s, 3H, CH_3_). ^13^C NMR (125 MHz, DMSO-*d_6_): δ* 157.6, 154.3, 148.8, 146.6, 144.4, 136.2, 133.3, 133.2, 131.1, 129.7, 129.6, 129.0, 128.9, 128.8, 128.2, 127.7, 127.6, 127.4, 127.2, 124.1, 39.9. HR EIMS: *m*/*z* calcd for C_22_H_18_N_4_O_4_S_2_ [M]^+^ 466.5335; Found: 466.5320.

##### 4.2.17. N′-(4-([1,1′-biphenyl]-4-yl)thiazol-2-yl)-4-chlorobenzenesulfonohydrazide (**17**)

Yield: 70%; m.p: 167-169 °C. ^1^H NMR (500 MHz, DMSO-*d_6_): δ* 11.50 (s, 1H, NH), 11.35 (s, 1H, NH), 8.64 (d, *J* = 8.2 Hz, 2H, Ar-H), 8.40 (d, *J* = 7.2 Hz, 2H, Ar-H), 8.16 (d, *J* = 8.9 Hz, 2H, Ar-H), 8.10 (d, *J* = 8.5 Hz, 2H, Ar-H), 7.80–7.89 (m, 1H, Ar-H), 7.63 (d, *J* = 8.4 Hz, 2H, Ar-H), 7.26 (dd, *J* = 6.0, 2.3 Hz, 1H, Ar-H), 7.09 (s, 1H, CH). ^13^C NMR (125 MHz, DMSO-*d_6_): δ* 191.7, 171.9, 169.8, 165.4, 162.9, 148.2, 148.0, 147.8, 147.0, 146.0, 135.3, 135.2, 134.9, 134.0, 132.4, 130.7, 130.7, 130.5, 128.8, 128.5, 128.3. HR EIMS: *m*/*z* calcd for C_21_H_16_ClN_3_O_2_S_2_ [M]^+^ 441.9578; Found: 441.9561. 

##### 4.2.18. 2,5-Dinitro-N′-(4-(3-nitrophenyl)thiazol-2-yl)benzenesulfonohydrazide (**18**)

Yield: 80%; m.p: 189–193 °C. ^1^H NMR (500 MHz, DMSO-*d_6_): δ* 11.41 (s, 1H, NH), 11.25 (s, 1H, NH), 8.13 (s, 1H, Ar-H), 8.06 (dd, *J* = 6.0, 1.7 Hz, 1H, Ar-H), 7.94 (dd, *J* = 8.0, 2.3 Hz, 1H, Ar-H), 7.51 (dd, *J* = 7.5, 3.9 Hz, 1H, Ar-H), 7.49 (t, *J* = 7.1 Hz, 1H, Ar-H), 7.10 (m, 1H, Ar-H), 2.46 (s, 1H, CH). ^13^C NMR (125 MHz, DMSO-*d_6_): δ* 192.3, 172.0, 169.8, 148.2, 146.0, 145.2, 138.7, 135.3, 135.2, 132.6, 130.7, 129.6, 129.1, 129.0, 128.5. HR EIMS: *m*/*z* calcd for C_15_H_10_N_6_O_8_S_2_[M]^+^ 466.4052; Found: 466.4038.

##### 4.2.19. N′-(4-([1,1′-biphenyl]-4-yl)thiazol-2-yl)-2,5-dinitrobenzenesulfonohydrazide (**19**)

Yield: 76%; m.p: 197–200 °C. ^1^H NMR (500 MHz, DMSO-*d_6_): δ* 11.38 (s, 1H, NH), 11.24 (s, 1H, NH), 8.26 (s, 1H, Ar-H), 8.03 (d, *J* = 8.2 Hz, 2H, Ar-H), 7.87 (dd, *J* = 8.2, 2.4 Hz, 1H, Ar-H), 7.67 (d, *J* = 8.4 Hz, 1H, Ar-H), 7.62 (d, *J* = 8.5 Hz, 2H, Ar-H), 7.62 (d, *J* = 7.2 Hz, 1H, Ar-H), 7.40–7.49 (m, 1H, Ar-H), 7.34 (dd, *J* = 7.0, 3.8 Hz, 1H, Ar-H), 2.11 (s, 1H, CH). ^13^C NMR (125 MHz, DMSO-*d_6_): δ* 161.7, 159.3, 158.8, 153.7, 148.0, 147.3, 141.5, 135.3, 134.9, 130.8, 128.9, 128.0, 126.9, 123.7, 123.3, 122.1, 66.5, 48.4, 39.8, 23.2, 20.2. HR EIMS: *m*/*z* calcd for C_21_H_15_N_5_O_6_S_2_ [M]^+^ 497.5078; Found: 497.5065.

##### 4.2.20. 4-Nitro-N′-(4-(3-nitrophenyl)thiazol-2-yl)benzenesulfonohydrazide (**20**)

Yield: 83%; m.p: 219–222 °C. ^1^H NMR (500 MHz, DMSO-*d_6_): δ* 11.4 (s, 1H, NH), 8.25 (s, 1H, NH), 8.23 (dd, *J* = 8.7, 1.9 Hz, 1H, Ar-H), 8.04 (dd, *J* = 6.4, 1.8 Hz, 1H, Ar-H), 7.84 (d, *J* = 6.4 Hz, 2H, Ar-H), 7.74 (d, *J* = 7.3 Hz, 2H, Ar-H), 7.41–7.44 (t, 1H, Ar-H), 3.50 (s, 1H, CH). ^13^C NMR (125 MHz, DMSO-*d_6_): δ* 192.3, 175.3, 169.9, 153.7, 147.3, 145.1, 138.6, 132.6, 129.0, 128.8, 128.5, 128.4, 127.0, 126.9, 123.3. HR EIMS: *m*/*z* calcd for C_15_H_11_N_5_O_6_S_2_ [M]^+^ 421.4080; Found: 421.4072.

##### 4.2.21. N′-(4-([1,1′-biphenyl]-4-yl)thiazol-2-yl)-4-nitrobenzenesulfonohydrazide (**21**)

Yield: 74%; m.p: 178–181 °C. ^1^H NMR (500 MHz, DMSO-*d_6_): δ* 11.62 (s, 1H, NH), 11.44 (s, 1H, NH), 8.64 (d, *J* = 8.2 Hz, 2H, Ar-H), 8.40 (d, *J* = 7.2 Hz, 2H, Ar-H), 8.10 (d, *J* = 8.5 Hz, 2H, Ar-H), 8.01 (d, *J* = 8.9 Hz, 2H, Ar-H), 7.80–7.89 (m, 1H, Ar-H), 7.99 (d, *J* = 8.4 Hz, 2H, Ar-H), 7.26 (dd, *J* = 6.0, 2.3 Hz, 1H, Ar-H), 2.89 (s, 1H, CH). ^13^C NMR (125 MHz, DMSO-*d_6_): δ* 191.6, 171.8, 169.7, 165.3, 162.8, 148.1, 148.2, 147.7, 147.1, 146.3, 135.2, 135.4, 134.8, 134.0, 132.3, 130.6, 130.6, 130.4, 128.7, 128.4, 128.2. HR EIMS: *m*/*z* calcd for C_21_H_16_N_4_O_4_S_2_ [M]^+^ 452.5045; Found: 452.5028.

### 4.3. Molecular Docking Study Assay

To understand the binding mode of synthesized compounds against both the targeted enzymes, acetylcholinesterase (AChE) and butyrylcholinesterase (BuChE), a molecular docking study was conducted using the Molecular Operating Environment (MOE) software package to corroborate the in vitro and in silico results well. The PDB codes 1ACL for AChE and 1P0P for BChE were used to retrieve the crystal structures of both targets from the RCSB protein databank. The crystallographic structures and all synthesized compounds were protonated, and energy was minimized using the default MOE-Dock module parameters, resulting in optimized enzyme and compound structures. These improved enzyme and chemical structures were then used for the docking study. Our prior investigations [57,58] include detailed descriptions of the docking protocol.

### 4.4. Acetylcholinesterase and Butyrylcholinesterase Activity Assay Protocol 

The assay for acetylcholinesterase and butyrylcholinesterase inhibitory potential was carried out according to the Ellman et al., method with slight modification [59]. The reaction mixture had a total volume of 100 µL. It comprised 60 µL of Na_2_HPO_4_ buffer with a concentration of 50 mM and a pH of 7.7. In total, 10 µL of test compound (well-1) with a concentration of 0.5 mM was added, followed by the addition of 10 µL (0.005 unit well-1) of an enzyme. The substances were mixed and pre-read at 405 nm. Then, the substances were pre-incubated at 37 °C for 10 min. The reaction was started by the addition of 10 µL of 0.5 mM well-1 substrate (acetylthiocholine iodide/butyrylthiochloine chloride), followed by the addition of 10 µL DTNB (0.5 mM well-1). Absorbance was measured at 405 nm after 15 min of incubation at 37 °C by using a 96-well plate reader Synergy HT, BioTek, USA. All experiments were performed with their respective controls in triplicate. Donepezil was used as a standard drug. The % inhibition was computed using the equation below.
Inhibition (%) = Control − Test/control × 100

Control EZ-Fit Enzyme kinetics software (Perrella Scientific Inc. Amherst, USA) was used for the calculation of IC_50_ values.

### 4.5. Statistical Analysis

All of the measurements were taken in triplicate, and Microsoft Excel 2003 was used to conduct the statistical analysis. The results are shown as standard error means (SEM).

## Data Availability

Not applicable.

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
