# Peer review of "Synthesis, DFT Studies, Molecular Docking and Biological Activity Evaluation of Thiazole-Sulfonamide Derivatives as Potent Alzheimer’s Inhibitors"

_molecules, 2023, doi:10.3390/molecules28020559_

Round 1

Reviewer 1 Report

S. Khan and his group reported the Synthesis, DFT studies, Molecular Docking, and Biological Activity Evaluation of Thiazole-Sulfonamide derivatives as Potent 3 Anti-Alzheimer’s Inhibitors. Experimental data supported the results. some of the prepared compounds were potent. The manuscript is nicely set up, and the language of the manuscript is good, however, some minor editing is required There are some comments below authors need to address before publication

Major

1.         The authors claim that (The purity of the synthesized compounds was achieved by washing with n-hexane) which is not convincing and not common in organic synthesis. So, the authors must provide the NMR spectrum in the supplementary data to check the purity.

2.         It would be better if the authors discussed the characterization of the prepared compounds in the discussion part (chemistry). This would be more interesting for organic chemistry readers.

3.         Authors should provide references for the synthetic method as this method was reported previously

4.         In the experimental section. If any of the prepared compounds were reported before, authors should cite the ref,

6.         Molecular docking study and DFT calculations are poorly described. Authors should compare with the standard drug.

Minors

7.      In line 56 change the word deceases to decreases.

8.     The manuscript's language is good; however, some minor editing is required.

9.     Reaction conditions would be better if written as a caption in scheme1

Author Response

Reviewer-1: Comments and Suggestions for Authors S. Khan and his group reported the Synthesis, DFT studies, Molecular docking, and Biological Activity Evaluation of Thiazole-Sulfonamide derivatives as Potent 3 Anti-Alzheimer’s Inhibitors. Experimental data supported the results. Some of the prepared compounds were potent. The manuscript is nicely set up, and the language of the manuscript is good, however, some minor editing is required There are some comments below authors need to address before publication Major 

1. The authors claim that (The purity of the synthesized compounds was achieved by washing with n-hexane) which is not convincing and not common in organic synthesis. So, the authors must provide the NMR spectrum in the supplementary data to check the purity.
Reply: NMR spectrum added in the supplementary information according to the kind reviewer suggestion. 

2. It would be better if the authors discussed the characterization of the prepared compounds in the discussion part (chemistry). This would be more interesting for organic chemistry readers. 
Reply: Incorporated according to the kind reviewer suggestion.

3. Authors should provide references for the synthetic method as this method was reported previously. 
Reply: Reference incorporated where necessary according to the suggestion of kind reviewer. 

4. In the experimental section. If any of the prepared compounds were reported before, authors should cite the ref, 
Reply: Reference incorporated where necessary according to the suggestion of kind reviewer. 

5. Molecular docking study and DFT calculations are poorly described. Authors should compare with the standard drug. 
Reply: Changes have been incorporated according to the suggestion of kind reviewer. Minors 

6. In line 56 change the word deceases to decreases. 
Reply: Corrected according to the suggestion of kind reviewer. 

7. The manuscript's language is good; however, some minor editing is required. 
Reply: Corrected according to the suggestion of kind reviewer. 

8. Reaction conditions would be better if written as a caption in scheme1 
Reply: Incorporated in caption according to the kind reviewer suggestion.

Reviewer 2 Report

Dear authors,

Congratulations for all the hard work in lab and computational studies. Nevertheless, in my scientific opinion there are several corrections or modifications that could be done in order this article to have the high standards of the journal to be publihed and be openly accessed.

- Introduction should be re-written carefully and checked by someone who is native or professional english speaker. There are scientific and language errors that do not give the best impression. For example, penicillin does not contain thiazole nucleus. Moreover, acetylcholine is hydrolised into choline AND acetic acid. The language errors are important also because there are mistakes between in singluar and plural forms that any professional or native english speaker would immediately notice.

-In Chemistry section again plural forms of reagents should be also checked.

- In the beginning of Experimental part 4.1, i think that in text the NH4SCN step is missing.

- Experimental procedures of each step in Chemistry section should be explained thoroughly and not summed up.

- In Conclusion please check singular and plural forms again for language.

- SAR section needs more analysis because the are more comparisons to do regarding the substituents. You could say more abour how in your opinion the nature and size of the substituents  potentially have these particular activity effects on the final products.

Kind regards

Author Response

Reviewer-2:

Comments and Suggestions for Authors

Dear authors,

Congratulations for all the hard work in lab and computational studies. Nevertheless, in my scientific opinion there are several corrections or modifications that could be done in order this article to have the high standards of the journal to be published and be openly accessed.

- Introduction should be re-written carefully and checked by someone who is native or professional English speaker. There are scientific and language errors that do not give the best impression. For example, penicillin does not contain thiazole nucleus. Moreover, acetylcholine is hydrοlysed into choline AND acetic acid. The language errors are important also because there are mistakes between in singular and plural forms that any professional or native English speaker would immediately notice.

Reply: Corrected according to the suggestion of kind reviewer.

-In Chemistry section again plural forms of reagents should be also checked.

Reply: Corrected according to the suggestion of kind reviewer.

- In the beginning of Experimental part 4.1, i think that in text the NH4SCN step is missing.

Reply: According to the suggestion of kind reviewer, changes have been incorporated.

- Experimental procedures of each step in Chemistry section should be explained thoroughly and not summed up.

Reply: Now corrected and changes have been incorporated according to kind reviewer suggestion.

- In Conclusion please check singular and plural forms again for language.

Reply: According to the suggestion of kind reviewer, changes have been incorporated.

- SAR section needs more analysis because they are more comparisons to do regarding the substituents. You could say more about how in your opinion the nature and size of the substituents potentially have these particular activity effects on the final products.

Reply: Corrected and changes have been incorporated according to the suggestion of kind reviewer.

Round 2

Reviewer 1 Report

the manuscript has been improved. However, there are some English mistakes that should be corrected

I recommend publishing the current work after English editing

Author Response

Comments and Suggestions for Authors

The manuscript has been improved. However, there are some English mistakes that should be corrected. I recommend publishing the current work after English editing.

Reply: Corrected according to reviewer suggestion.